# Altered Insulin Clearance after Gastric Bypass and Sleeve Gastrectomy in the Fasting and Prandial Conditions

**DOI:** 10.3390/ijms23147667

**Published:** 2022-07-11

**Authors:** Marzieh Salehi, Ralph DeFronzo, Amalia Gastaldelli

**Affiliations:** 1Division of Diabetes, University of Texas Health at San Antonio, San Antonio, TX 78229, USA; defronzo@uthscsa.edu; 2South Texas Veteran Health Care System, Audie Murphy Hospital, San Antonio, TX 78229, USA; 3Cardiometabolic Risk Unit, CNR Institute of Clinical Physiology, 56124 Pisa, Italy

**Keywords:** insulin clearance, gastric-bypass surgery, sleeve gastrectomy, mixed meal test, hyperinsulinemic hypoglycemic clamp, insulin extraction

## Abstract

**Background**: The liver has the capacity to regulate glucose metabolism by altering the insulin clearance rate (ICR). The decreased fasting insulin concentrations and enhanced prandial hyperinsulinemia after Roux-en-Y gastric-bypass (GB) surgery and sleeve gastrectomy (SG) are well documented. Here, we investigated the effect of GB or SG on insulin kinetics in the fasting and fed states. **Method**: ICR was measured (i) during a mixed-meal test (MMT) in obese non-diabetic GB (n = 9) and SG (n = 7) subjects and (ii) during a MMT combined with a hyperinsulinemic hypoglycemic clamp in the same GB and SG subjects. Five BMI-matched and non-diabetic subjects served as age-matched non-operated controls (CN). **Results**: The enhanced ICR during the fasting state after GB and SC compared with CN (*p* < 0.05) was mainly attributed to augmented hepatic insulin clearance rather than non-liver organs. The dose-response slope of the total insulin extraction rate (InsExt) of exogenous insulin per circulatory insulin value was greater in the GB and SG subjects than in the CN subjects, despite the similar peripheral insulin sensitivity among the three groups. Compared to the SG or the CN subjects, the GB subjects had greater prandial insulin secretion (ISR), independent of glycemic levels. The larger post-meal ISR following GB compared with SG was associated with a greater InsExt until it reached a plateau, leading to a similar reduction in meal-induced ICR among the GB and SG subjects. **Conclusions**: GB and SG alter ICR in the presence or absence of meal stimulus. Further, altered ICR after bariatric surgery results from changes in hepatic insulin clearance and not from a change in peripheral insulin sensitivity.

## 1. Introduction

Roux-en-Y gastric-bypass (GB) surgery and sleeve gastrectomy (SG) induce a robust and durable improvement in glycemic control [1,2]. The glycemic improvement after GB and SG has been attributed to significant weight loss [3,4], enhancing insulin sensitivity and insulin clearance in proportion to the amount of weight loss [5,6,7].

We and others have demonstrated that fasting insulin clearance is enhanced in subjects with GB and SG compared to BMI- and age-matched controls [8,9] and compared to the same individual before their surgery [10,11,12]. Further, GB and, to lesser degree, SG, lead to improved glucose tolerance by enhancing prandial insulin levels [13,14,15,16]. Post-meal hyperinsulinemia several years after GB is caused by augmented insulin secretion and, possibly, a reduced insulin clearance rate (ICR) [17]. Furthermore, reduced prandial ICR after GB is exaggerated in those with post-GB hypoglycemia, which is suggestive of a pathogenic role for ICR [17].

In persons without gastrointestinal surgery, the liver has been implicated in the regulation of glucose metabolism through its ability to alter insulin clearance, independent of insulin sensitivity [18,19]. While much research has focused on the role of β-cell function in the beneficial glycemic effects of bariatric surgery, the physiological or pathophysiological changes in insulin kinetics in subjects with GB and SG [17] remain unknown.

In this study, we examined the effect of GB and SG on the extraction and clearance of exogenously administered insulin during glucose clamp in the fasting and fed condition. Furthermore, we compared insulin extraction and clearance rates (mainly by the liver) in relation to the increasing insulin concentrations produced endogenously during a mixed-meal test among non-diabetic patients with a history of GB and SG.

## 2. Results

### 2.1. Subjects

The GB, SG, and healthy controls (CN) had similar BMI, fat and lean body mass, age, HbA1c, and female-to-male ratios. The two surgical groups had similar pre-operative BMI, nadir body weight achieved in the first 12 months from surgery (75 ± 4 kg in both groups), as well as total weight loss and time since surgery (Table 1).

### 2.2. Insulin Kinetics during Hyperinsulinemic Glucose Clamp in the Fasting and Fed Conditions 

The glucose, glucose kinetics, and glucagon data were previously published as part of a report focusing on glucose counterregulatory response to insulin-induced hypoglycemia among GB and SG subjects [8]. The plasma concentrations of glucose at baseline were similar among the three groups and decreased rapidly to the target (60 mg/dL) with the infusion of insulin and were maintained steady at this level throughout the study (Table 2). The baseline plasma insulin concentrations were similar among the surgical and non-surgical subjects (Figure 1b). A steady-state plateau of hyperinsulinemia was achieved in all three groups, with lower levels in the GB and SG compared to the CN (*p* < 0.05) (Figure 1b). The fasting ISR values did not differ between the three groups (Table 2; Figure 1d). The fasting ICR was greater in both surgical groups compared to the controls (Table 2; Figure 1a), which was mainly accounted for by the higher hepatic insulin clearance (*p* < 0.01; Table 2) [20].

With a glycemic reduction from the baseline during the clamp, the endogenous insulin secretion rate (ISR) declined in three groups (*p* < 0.001; Figure 1d). The glucose infusion rate (M) required to maintain the premeal glycemic target (60 mg/dL) and peripheral insulin sensitivity calculated by the glucose infusion rate divided by the steady-state plasma insulin concentration (M/I) was similar in the three groups (Table 2).

The induced hyperinsulinemia reduced ICR in all three groups, reaching a plateau at 60 min, but the ICR remained greater in the surgical subjects compared to the CN for the remainder of the clamp study (Figure 1a). As a result, the circulatory levels of insulin were significantly lower in the surgical subjects compared to the CN (Figure 1b). The average slope of each subject’s plot of total insulin extraction rate versus the systemic insulin concentrations was significantly smaller in the CN compared to the GB and SG subjects (GB: 0.42 ± 0.03, SG: 0.49 ± 0.06, CN: 0.35 ± 0.03 (pmol·min^−1^·m^−2^)/(pmol/L); *p* < 0.05; Figure 2a).

After meal ingestion, to maintain the plasma glucose at the target, the glucose infusion rates had to be reduced, but then gradually increased at different rates between the surgical and nonsurgical subjects as a result of the variations in glucose influx secondary to GI surgeries. Following meal consumption, the β-cell secretion increased in the first 30–60 min in the GB subjects without any significant change in insulin secretory response in the SG subjects or the non-surgical controls (Figure 1d).

The meal ingestion during the hyperinsulinemic hypoglycemic clamp led to a similar brief reduction in plasma insulin levels in all three groups; the mean absolute declines in plasma insulin concentrations from 120 min (premeal) to 140 min (postmeal nadir level) were 225 ± 60, 164 ± 128, and 184 ± 90 pmol/L for the GB, SG, and CN, respectively; *p* < 0.01; Figure 1b). Within 60 min, the plasma insulin concentration returned to pre-meal values. In parallel with the short-lived insulin reduction after the meal ingestion, the total insulin extraction rates increased briefly in the GB subjects but decreased in the SG and CON (*p* < 0.05) before returning to premeal values within 60 min (Figure 1c and Figure 2a). As a result, the AUC_ICR_ was increased in the GB subjects for 3 h after meal ingestion compared to the SG subjects in this setting (Figure 1a, inset).

The fasting hepatic insulin clearance was inversely associated with the total fat mass (ρ = −0.5, *p* < 0.05); there was no correlation between the fasting hepatic insulin clearance and the peripheral insulin sensitivity, calculated as M/I. The ICR during the steady-state period of clamp was correlated with the M/I (ρ = 0.5, *p* < 0.05).

### 2.3. Glucose Response, Insulin Secretion, and Insulin Kinetics during Mixed-Meal Test (MMT)

During MMT, despite having similar fasting glucose and AUC_Glucose-3h_ (i.e., 0–180 min), the GB subjects had a larger peak glucose and glucose excursion (peak–nadir) as well as a greater AUC_Glucose-1 h_ (i.e., 0–60 min) compared to the SG subjects (*p* < 0.05, Table 3).

The insulin secretory responses to meal ingestion (AUC_ISR-1h_ and AUC_ISR-3h_) were significantly greater in the GB- than in the SG-treated subjects (*p* < 0.05, Table 3). The fasting insulin levels were similar among the GB and SG subjects but, in parallel with the altered glycemic pattern, the increase in early insulin concentration (AUC_Insulin-1h_) was larger in the GB compared to the SG (*p* < 0.05). The time to reach the peak insulin concentration was significantly longer than that taken to reach the maximum ISR values (*p* < 0.0001), although it was not significantly different between the GB and SG subjects.

The fasting ICR was similar among the surgical groups (Table 3; Figure 3a). Following meal ingestion, ICR diminished, reaching nadir values (in 20–50 min) that were similar in the GB and SG (Figure 3a). The increased plasma insulin response in the GB versus the SG (Table 3) was entirely accounted for by an increase in the insulin secretory response to meal ingestion (Figure 3b). As the insulin levels rose in the first part of the MMT (Figure 2b, solid arrow), the extraction of endogenous insulin rose, reaching its maximum at 10 min, and then plateaued, but at a much higher level in the GB compared to the SG. In the second part of the MMT, and in parallel with the decline in the plasma insulin levels from peak to premeal values (Figure 2b, dashed arrow), the InsExt decreased. The levels of fractional hepatic insulin extraction among the surgical groups were similar (Table 3). There was no correlation between the fasting or prandial ICR and the peripheral tissue (which is primarily composed of muscle) insulin sensitivity measured during the insulin clamp. The ICR parameters were not associated with any other glycemic or insulin secretory profile during MMT.

## 3. Discussion

Here, we demonstrate that enhanced ICR after bariatric surgery in the fasting state is attributable to variations in insulin clearance in the liver. Further, greater hepatic insulin clearance after GB and SG is not associated with better peripheral insulin sensitivity. In the absence of a meal stimulus, raising the systemic insulin concentration to prandial levels typically observed after GB by exogenous insulin administration lowers the ICR in both surgical and non-surgical subjects. The major finding in the present study is that the total extraction of exogenous insulin was much larger in GB and SG compared to CN subjects across the range of systemic insulin that is typically observed in the prandial setting after GB. We also found that during MMT, increases in endogenous insulin secretion lead to identical reductions in ICR in GB and SG subjects despite a much larger prandial nutrient flux and higher insulin levels after GB. This phenomenon is most plausibly explained by the larger capacity of the liver after GB to extract presented endogenous insulin at much higher levels than after SG before it reaches saturation. Collectively, these data suggest that the regulation of insulin clearance in the prandial and fasting states is altered after bariatric surgery.

Fasting plasma concentrations of insulin decrease dramatically within 1–4 weeks after GB or SG before any substantial changes in body weight or plasma C-peptide concentrations occur, suggesting an increase in endogenous insulin clearance [11,12,21,22,23]. This is consistent with our findings, which demonstrate that the increased basal ICR in GB and SG patients persists for many years after their surgery. It is well recognized that weight loss, whether induced by life-style interventions or bariatric surgery, improves insulin sensitivity and fasting insulin clearance [5,6,7]. Studies comparing the effect of equal weight loss achieved by GB or calorie restriction alone on fasting insulin clearance 2–4 weeks after surgery have found that GB has no additional benefit over calorie restriction on fasting insulin kinetics [24,25,26]. However, despite similar increases in basal ICR, the ICR calculated during exogenous insulin infusion (euglycemic hyperinsulinemic clamp) has been reported to be larger after GB compared to dietary restriction alone at similar weight loss [23,25]. These weight-loss-independent beneficial effects of GB over calorie restriction on the clearance of intravenously infused insulin after surgery were not related to changes in peripheral insulin sensitivity (M/I), which remained unchanged [23]. These data support the recent debate on whether the liver has the capacity to regulate glucose metabolism by altering the metabolic clearance rate of insulin independent of insulin sensitivity [18,27]. In the present study, we used a novel method to allow us to estimate time changes in insulin clearance. Considering that peripheral insulin clearance (primarily in the kidneys and, to a lesser extent, the muscles) is constant over a wide range of plasma insulin concentrations [28,29] and that the three groups had similar degrees of peripheral insulin sensitivity (M/I), we can assume that the groups had similar peripheral insulin clearance (since it is a function of sensitivity) and that the significant difference in basal ICR among the surgical and matched non-surgical groups was mainly due to greater hepatic insulin clearance after these surgeries. Thus, our findings support a primary role for the liver in the alteration in insulin clearance after GB or SG, independent of changes in peripheral insulin sensitivity. The underlying mechanisms by which bariatric surgery alters fasting hepatic insulin clearance remains to be investigated.

To examine ICR when insulin is presented to the liver by the portal route (endogenous β-cell release) compared with the peripheral route (exogenous inulin infusion), we used MTT alone and combined with a hyperinsulinemic glucose clamp. Our clamp study allowed us to evaluate the relationship between total insulin extraction and induced hyperinsulinemia that is typical of prandial insulin levels after GB (~1500 pmol/min), in the absence and presence of meal stimuli, while insulin secretion is maximally suppressed.

In response to the increasing exogenous insulin delivery to the liver (hyperinsulinemic clamp), the ICR declined in all the subjects, but the steady-state insulin levels during hyperinsulinemia were much lower after GB and SG than in the controls, even though the whole-body (primarily reflects muscle) insulin sensitivity (M/I) was comparable across all three groups. These results are consistent with previous studies showing that gastric bypass increases the insulin clearance rate, independent of changes in insulin sensitivity [23,25]. Here, we add to this knowledge by demonstrating that the total insulin extraction of intravenously infused insulin (InsExt) is increased in both GB- and SG-treated subjects compared to BMI- and age-matched controls, leading to lower steady-state insulin values.

Prandial hyperinsulinemia several years after GB, particularly in those with GB-related hypoglycemia, has been attributed to both exaggerated β-cell output and possibly reduced insulin clearance [17,30]. The augmented insulin response when glucose is administered orally as opposed to intravenously, known as the incretin effect, has been attributed to both increased insulin secretion and decreased insulin clearance in healthy individuals [31,32,33,34,35]. The changes in insulin clearance during nutrient ingestion are attributed almost exclusively to the liver, considering that peripheral insulin clearance does not change [28,29]. The mechanism through which the oral ingestion of nutrients or glucose decreases insulin clearance is largely unknown, but ICR reduction seems to be associated with enhanced insulin secretion, and both are related to the size [36,37,38] and the composition of ingested nutrient [39]. Therefore, it has been speculated that the reduction in ICR after glucose ingestion is mainly determined by the insulin load delivered to the liver, and that this process is saturable [27,40].

The prandial reduction in ICR during the meal studies in our experiment did not differ between the GB and SG subjects despite the two-fold difference in mean insulin secretory response between the two groups. Within the first 10–20 min of the meal ingestion, when the amount of insulin reaching the liver was significantly increased, the extraction of endogenous insulin (mainly by the liver) also increased significantly, reaching peak values in both the GB and the SG subjects; however, the total insulin extraction was much larger in the GB compared to the SG subjects. After 10–20 min, the total amount of insulin removed by the liver plateaued while the plasma insulin concentrations continued to increase, particularly in those with GB. This was reflected by the longer time that it took the GB subjects to reach peak plasma insulin concentrations. Altogether, our results indicate that at the prandial insulin concentrations typically seen after GB, the liver has the capacity to remove the presented insulin load shortly after meal ingestion.

The effect of meal ingestion on insulin clearance was also examined during the fixed hyperinsulinemia induced by exogenous insulin infusion (clamp). In this setting, when the InsExt had reached the steady state (110–120 min), meal ingestion had minimal further influence on insulin clearance. However, we observed a small but significant prandial decline in insulin associated with a brief increase in InsExt, particularly in the GB subjects, whose ISR also increased, while there were minimal changes in ISR or InsExt in the SG and CN subjects. As a result, the prandial areas under the curve of the ICR over the premeal values were larger in the GB subjects compared to the SG subjects. The early ICR-enhancing effects of meal ingestion in this setting appear to be at odds with the overall prandial ICR reduction reported in the present and previous studies [17,35]. However, in our study, the increase in prandial InsExt in the first 10–20 min was shared during meal studies with and without hyperinsulinemia induced by exogenous insulin infusion. Therefore, it is plausible that there is a temporal InsExt/ICR response to meal ingestion.

We did not measure the portal insulin concentration or liver blood flow, which can also influence the hepatic extraction of insulin [41,42]. However, it is plausible that an increase in the portosystemic insulin gradient after meal ingestion during clamp in proportion to the increase in nutrient flux, particularly in the GB subjects, resulted in increased hepatic insulin uptake under fixed hyperinsulinemia—created by the exogenous insulin infusion. The exaggerated hepatic arteriovenous insulin gradient in a dog model was shown to increase ICR [42]. It also is possible that nutrient sensory signals in the portal system influence insulin clearance. The ingestion of amino acid compared to intravenous amino acid infusion was shown not only to increase ISR but also to augment insulin clearance, leading to insulin levels that were lower than expected based on the increased insulin secretory response [43]. Finally, hepatic zonation, based on the proximity of hepatocytes to the portal triad compared to the central vein, has been shown to influence the metabolic function of hepatocytes in response to hormones or nutrients [44]. In our study, the slope of the increase in the InsExt during the increase in plasma insulin concentration created by exogenous insulin infusion was much smaller when the insulin was presented exogenously than when it was presented endogenously (Figure 2). It is unclear whether insulin is distributed to different hepatocytes, which have different capacities for insulin uptake and degradation, if it is presented by the hepatic artery or by the portal vein.

There are limitations to our study that merit attention. As noted earlier, our study design was limited to addressing the effects of portal insulin, glucose concentration, or hypoglycemia on insulin clearance. However, it has been previously shown that hepatic insulin extraction in dogs is not affected by tolbutamide-induced hypoglycemia [45]. Furthermore, the number of subjects in each of the groups was relatively small, which may have reduced our power to detect differences in some of the outcomes of interest, such as peripheral insulin sensitivity, among the surgical and non-surgical subjects. However, despite the small sample size, the difference in insulin clearance, mainly by the liver, among the surgical and non-surgical groups was consistent. Further, we did not find any correlations between the peripheral insulin sensitivity and the hepatic insulin clearance in each subject. 

## 4. Materials and Methods

### 4.1. Subjects

The subjects represent the group who participated in a previously published study intended to investigate glucose counterregulatory response to insulin-induced hypoglycemia among patients with GB or SG [8]. Nine subjects with previous GB were enrolled in order of their response to advertisement or clinical visits. Seven subjects with prior history of SG and 5 healthy control subjects without prior GI surgery (CN) were also recruited to match for BMI, age, and fat/lean mass with GB subjects. The surgical subjects were recruited at least 2 years after their bariatric surgery. All subjects were weight-stable for at least 3 months prior to study. The control subjects had no personal or family history of diabetes and they had a normal oral glucose tolerance test (2-h plasma glucose level <7.8 mmol/L following 75-gram-oral-glucose-tolerance test). All subjects were free of diabetes, gastrointestinal disease, and renal and liver dysfunction, and none took any medications that interfere with glucose metabolism for at least one week prior to study. The Institutional Review Board of the University of Texas Health at San Antonio approved the protocol, and all participants provided written informed consent before participating in any experiments.

### 4.2. Experimental Protocols

Participants were instructed not to engage in excessive physical activity and maintain 150–200 g carbohydrate ingestion for 3 days before each visit. Participants were admitted in the morning after a 10-h overnight fast at the Bartter Research Unit at Audie Murphy VA Hospital on two separate days.

Body composition was assessed using dual-energy X-ray absorptiometry and waist circumference was measured. Intravenous (IV) catheters were placed in an antecubital vein for the withdrawal of blood and the infusion of insulin and glucose. The IV site used for blood sampling was continuously warmed with a heating pad.

*MMT:* After drawing fasting blood samples, at 0 min, surgical subjects consumed a 237-milliliter liquid mixed meal containing 350 kcal with 57% carbohydrate, 15% protein, and 28% fat (Ensure Plus, Abbott Laboratories, Abbott Park, IL, USA) within 10 min. Blood samples were taken from 0 min to 180 min, and plasma was separated within 60 min for storage at −80 °C until assay. Plasma glucose concentrations were determined at the bedside using a glucose analyzer.

*Hyperinsulinemic hypoglycemic clamp combined with mixed meal ingestion*: After withdrawal of fasting blood at 0 min, a primed and continuous infusion of recombinant human insulin (Humulin 100 U/mL) diluted in isotonic saline and mixed with 2 mL of subjects’ blood was started and continued at 120 mU/min/m^2^ for the duration of the study. Blood was sampled at 5–10 min intervals and a variable infusion of 20% glucose was infused to maintain the plasma glucose at a target of 55–60 mg/dL (~3 mmol/L). At 120 min, subjects consumed a 140-mL liquid mixed meal containing 33 g whey protein, 14 g glucose, and 12.7 g corn oil in a semi-recumbent position. The meal was consumed over 10 min and the glucose clamp was maintained by a variable glucose infusion for 3 h. Throughout the study, blood samples were collected at timed intervals and stored on ice; plasma was separated within 60 min and stored at −80 °C until assay.

### 4.3. Assays

Blood samples were collected in heparin for measurement of insulin and glucose and in aprotinin/heparin/EDTA for assay of C-peptide [17,46]. Plasma glucose was determined using Analox GM9 Glucose Analyzer (Analox Instruments, Stourbridge, UK). Insulin (DIAsource, Neuve, Belgium) and C-peptide (Millipore, Billerica, MA, USA) were measured by commercial radioimmunoassay.

### 4.4. Modeling Analysis

Insulin secretion rates (ISR) were derived from plasma C-peptide concentrations by deconvolution using the Van Cauter two-pool model (Figure 4a) with population estimates of model parameters [30]. Post-GB surgery is associated with fast insulin response with very high insulin and C-peptide concentration increases and falls that often result in low and even negative values of ISR during the second phase of insulin secretion. Considering that C-peptide clearance is mainly renal, and that it should not be affected by GB, we hypothesized that model error might be due to an incorrect model estimate of fractional transfer rates between fast and slow compartments of C-peptide kinetics. Thus, we estimated that the exchange rate from the fast to slow compartment (k21, Figure 4a) could be up to 60% less in GB subjects, with an additional reduction of 30% in the exchange from slow to fast compartments (k12, Figure 4a) compared to the population values calculated using the formula proposed by Van Cauter [47]. The new values were derived from previous data recently published [48,49].

Insulin sensitivity (M/I) during clamp was computed as the average glucose infusion rate from 110–120 min divided by mean plasma insulin concentration over the same period [50].

**Figure 4 ijms-23-07667-f004:**
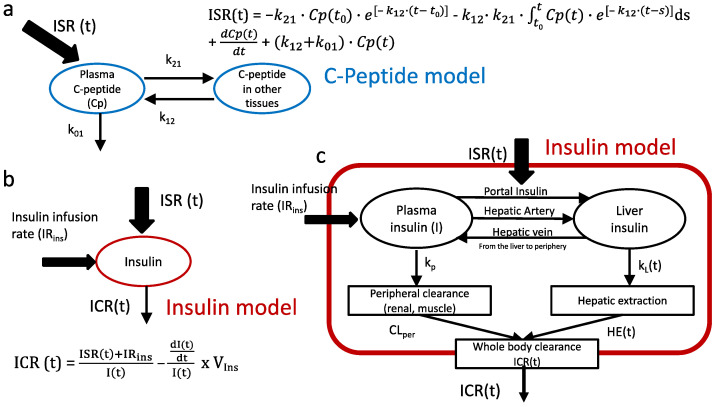
(**a**) C-peptide model according to Van Cauter [30,47] and derived equation for the estimation of pre-hepatic insulin secretion rates (ISR) from C-peptide concentrations measured over time, from Cp(t), and from k values (fractional clearance rates between compartments). We estimated that k21 could be up to 60% less in GB subjects, with an additional reduction of 30% in k12 compared to the population values calculated using the formula proposed by Van Cauter [47]. (**b**) Insulin model for the calculation of whole-body clearance rate (ICR) [18,51]. (**c**) Insulin is mainly cleared by the liver (more than 60% during first pass and also during second pass) and in a small, fixed proportion by peripheral tissues (kidneys and muscles). Thus, any temporal change observed in insulin clearance reflects a change in hepatic insulin clearance.

Metabolic insulin clearance rate (ICR) in the fasting state was calculated as fasting ISR (ISR(t_0_)) divided by fasting plasma insulin concentration (I(t_0_)), since steady-state insulin extraction (InsExt) equals ISR [18].
Fasting:    ICR(t_0_) = ISR(t_0_)/I(t_0_)(1)

During meal studies, a single-pool model to describe insulin kinetics (Figure 4b) was used as previously reported [18]. The rate of total insulin extraction, InsExt (t), i.e., liver plus peripheral tissues, and ICR(t) at each time point during meal studies and hyperinsulinemic clamp were calculated as previously described by assuming the insulin kinetics as single-pool model [18,52], i.e., during the MTT:(2)$$\mathrm{MTT}:\;\mathrm{InsExt}\;\left(t\right)\;=\;\mathrm{ISR}\left(t\right)-\frac{\frac{\mathrm{dI}\left(t\right)}{\mathrm{dt}}}{I\left(t\right)}\;\times \;V_{\mathrm{Ins}}\;(\mathrm{pmol}\cdot \min^{-1}\cdot m^{-2})$$
(3)$$\mathrm{MTT}:\;\mathrm{ICR}\;\left(t\right)=\frac{\mathrm{InsExt}\left(t\right)}{I\left(t\right)}=\;\frac{\mathrm{ISR}\left(t\right)}{I\left(t\right)}-\frac{\frac{\mathrm{dI}\left(t\right)}{\mathrm{dt}}}{I\left(t\right)}\;\times \;V_{\mathrm{Ins}}\;(L\cdot \min^{-1}\cdot m^{-2})$$
where V_Ins_ is the volume of distribution of insulin (assumed constant), which is assumed to be 141 mL/kg [51,53,54].

During the hyperinsulinemic clamp, insulin extraction and clearance include the contribution of both exogenous and endogenous insulin.
Clamp: InsExt = Insulin infusion rates + ISR_clamp _ (pmol·min^−1^·m^−2^)(4)
(5a)$$\mathrm{Clamp}:\;{\mathrm{ICR}}_{\mathrm{exogenous}}\;=\frac{\mathrm{InsExt}}{I_{\mathrm{clamp}}-I_{\mathrm{end}}}=\frac{\;\mathrm{Insulin}\;\mathrm{infusion}\;\mathrm{rates}\;+{\;\mathrm{ISR}}_{\mathrm{clamp}}}{I_{\mathrm{clamp}}-I_{\mathrm{end}}}\;\left(L\cdot \min^{-1}\cdot m^{-2}\right)$$
(5b)$${\mathrm{ICR}}_{\mathrm{total}}\;=\frac{\mathrm{InsExt}}{I_{\mathrm{clamp}}}=\frac{\;\mathrm{Insulin}\;\mathrm{infusion}\;\mathrm{rates}\;+{\;\mathrm{ISR}}_{\mathrm{clamp}}}{I_{\mathrm{clamp}}}\;\left(L\cdot \min^{-1}\cdot m^{-2}\right)$$
where $I_{\mathrm{end}}$ is calculated as plasma insulin_clamp × C-peptide_clamp/fasting C-peptide, as previously described [28].

The insulin extraction is given by the sum of the peripheral insulin extraction rate k_p_, which is considered to be constant (according to [51,53,54] and the hepatic insulin extraction rate k_L_(t), which is time-dependent (Figure 4).

Considering that during MTT, total insulin clearance rates (ICR) include both liver and peripheral tissue, while during the clamp, the great majority of insulin is taken up from peripheral tissues, and only a minimal part is taken up by the liver during second pass, percentage hepatic insulin extraction (HE) can be estimated as previously described [55]:(6)$$\mathrm{HE}\left(\%\right)=\frac{\mathrm{ICR}\;\mathrm{MTT}-\;\mathrm{ICR}\;\mathrm{clamp}}{\mathrm{ICR}\;\mathrm{MTT}}$$

Given that insulin clearance in peripheral tissues is assumed to be related only to the degree of insulin resistance [51,53,54], all changes observed during insulin infusion (or meal test) compared to fasting values are due only to changes in hepatic clearance.
Hep-ICR (t) = ICRclamp (t) – ICR (t_0_) (L·min^−1^·m^−2^)(7)

Thus, hepatic ICR is modulated by the amount of insulin that reaches the liver, as well as leading to saturation in case of very high insulin levels or in liver disease [51].

### 4.5. Calculations and Analysis

Fasting plasma glucose, insulin, and C-peptide concentrations were computed as the average of the 3 samples drawn before the clamp or screening MTT, and the pre-meal values during clamp as the average of the samples drawn during the 10 min before the test meal (110–120 min). Glucose, insulin, and ISR responses to mixed-meal test were calculated as the incremental areas under the curve (AUC) over premeal values using the trapezoidal rule. AUC values for all prandial parameters were calculated for 0 min to 180 min after meal ingestion, as well as for 0 min to 60 min, to evaluate the early response, since GI surgeries alter prandial response patterns.

Mean ± SEM were computed for glucose, glucose infusion rate (M), insulin, ISR, and ICR from 110 min to 120 min (steady-state glycemia, before meal ingestion), 120–180 min, and 120–300 min (hypoglycemic clamp, after meal ingestion).

### 4.6. Statistical Analysis

Data are presented as mean ± SEM. MLAB software (MLAB, Civilized Software Inc., Silver Spring, MD, USA) was used for modeling analyses and the calculation of insulin secretion and clearance. The parameters of interest at baseline and during screening MTT or the hyperinsulinemic clamp were compared using X2 or ANOVA based on pre-specified comparisons among the groups (surgical vs. controls, and GB vs. SG). The effect of meal ingestion during the insulin clamp studies (time factor, before and after meal) and group effect (GB, SG, and CN) on studied outcomes were analyzed using repeated-measures ANOVA, using SPSS 26 (SPSS Inc., Chicago, IL, USA).

## 5. Conclusions

Our findings indicate that: (1) meal ingestion enhances insulin extraction (mainly by the liver) in proportion to altered nutrient flux after GB compared to SG and (2) the hepatic extraction of exogenously administrated insulin is increased after GB and SG.

## Figures and Tables

**Figure 1 ijms-23-07667-f001:**
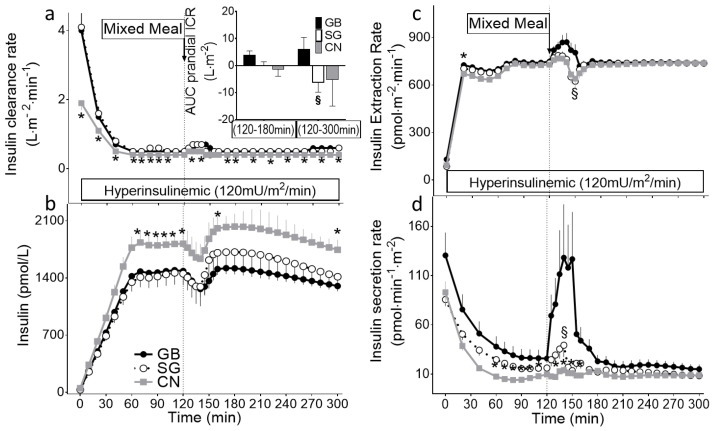
(**a**) Insulin clearance rate (ICR), (**b**) plasma insulin concentration, (**c**) insulin extraction rate, (**d**) and insulin secretion rate during hyperinsulinemic hypoglycemia clamp combined with mixed-meal ingestion in subjects with previous history of GB (solid black line, black bar) or SG (dashed line, white bar) or non-surgical controls (solid grey line, gray bar). AUC_ICR_ for the first hour (120 min to 180 min) and 3 h (120 min to 300 min) after meal ingestion are shown (inset). * *p* < 0.05 compared to GB and SG, § *p* < 0.05 compared to GB.

**Figure 2 ijms-23-07667-f002:**
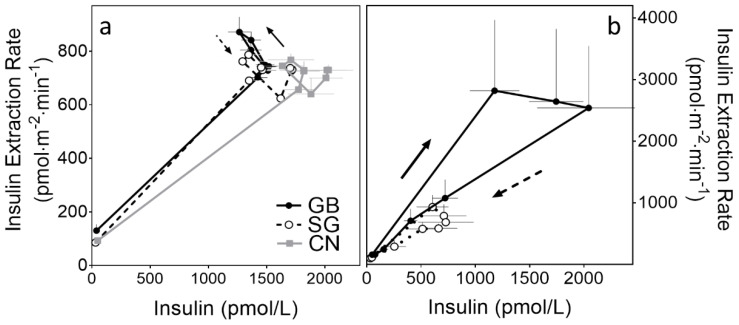
(**a**) The rates of insulin extraction in relation to increasing plasma insulin concentrations produced by exogenous insulin infusion during hyperinsulinemic hypoglycemic clamp with mixed-meal ingestion and (**b**) prandial endogenous insulin secretory response during mixed-meal test. Solid arrows represent the first phase of meal studies and dashed arrows the second part.

**Figure 3 ijms-23-07667-f003:**
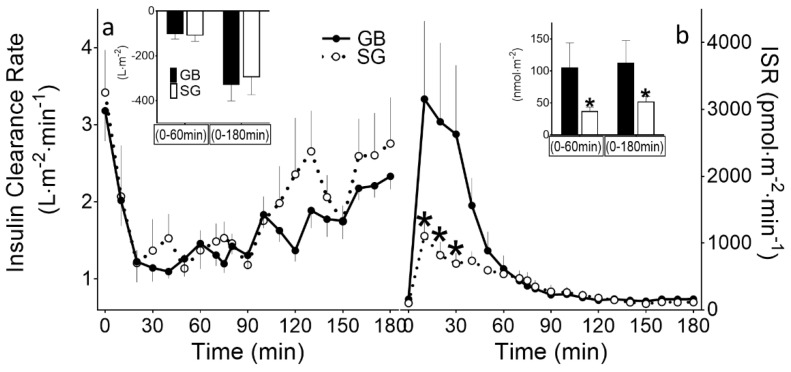
(**a**) Fasting and prandial clearance rates of insulin (ICR); (**b**) plasma insulin secretory response during mixed-meal test in subjects with previous history of GB (solid line, black bar) or SG (dashed line, white bar). Corresponding changes from baseline in AUC for 0 to 60 min and 0 to 180 min are shown (insets). * *p* < 0.05 compared to GB.

**Table 1 ijms-23-07667-t001:** Baseline characteristics of study subjects during the fasting condition in GB, SG, and CN groups.

	GB (9)	SG (7)	CN (5)
Age (years)	44.4 ± 4.0	48.3 ± 3.8	44.2 ± 4.3
BMI (kg/m^2^)	31.2 ± 2.1	32.3 ± 1.9	30.8 ± 2.9
Lean mass (kg)	53.1 ± 2.5	55.7 ± 4.6	46.8 ± 6.6
Fat mass (kg)	29.7 ± 3.3	29.9 ± 4.4	33.7 ± 4.8
Waist circumference (cm)	101.2 ± 5.6	98.6 ± 3.9	98.2 ± 5.8
Sex (M/F)	1/8	2/5	1/4
HbA1c (mmol/mol)	35 ± 1	34 ± 2	36 ± 1
HbA1c (%)	5.4 ± 0.1	5.3 ± 0.2	5.4 ± 0.1
Time since surgery (years)	5.3 ± 1.8	4.7 ± 0.5	
Preoperative BMI (kg/m^2^)	47.6 ± 2.0	46.6 ± 2.0	
Weight loss (kg)	45.0 ± 6.0	39.2 ± 5.6	
BMI loss (kg/m^2^)	16.3 ± 2.2	14.2 ± 1.6	
Max weight loss **^§^**	55.7 ± 3.8	52.5 ± 5.9	

Data are presented as mean ± SEM unless specified otherwise. GB, patients with prior history of gastric bypass; SG, sleeve gastrectomy patients; CN, non-surgical controls; **^§^** maximum weight loss at 6–12 months from surgery.

**Table 2 ijms-23-07667-t002:** Glucose and insulin kinetics during hypoglycemic hyperinsulinemic clamp before and after mixed-meal ingestion.

		GB (n = 9)	SG (n = 7)	CN (n = 5)
Glucose (mmol/L)	Basal	5.4 ± 0.1	5.2 ± 0.1	5.6 ± 0.1
	110–120 min	3.1 ± 0.0	3.2 ± 0.1	3.3 ± 0.0
GIR (µmol·min^−1^·kg^−1^)	110–120 min	21 ± 3	24 ± 4	17 ± 2
Insulin (pmol/L)	Basal	39 ± 7	29 ± 6	50 ± 8
	110–120 min	1491 ± 80	1458 ± 140	1819 ± 168 *
ISR (pmol·min^−1^·m^−2^)	Basal	131 ± 23	86 ± 7	93 ± 11
	110–120 min	26 ± 8	16 ± 2	7 ± 2 *
AUC_ISR_ (nmol·m^−2^)	120–180 min	2.5 ± 1.2	0.2 ± 0.3	0.4 ± 0.1 *
	120–300 min	1.4 ± 1.2	−0.5 ± 0.6	0.2 ± 0.0
Insulin clearance rate	Basal	3.5 ± 0.6	3.5 ± 0.7	1.9 ± 0.2 *
(L·min^−1^·m^−2^)	110–120 min	0.5 ± 0.0	0.5 ± 0.1	0.4 ± 0.1
Hepatic insulin clearance	Basal	3.3 ± 0.6	3.2 ± 0.7	1.7 ± 0.2 *
(L·min^−1^·m^−2^)	110–120 min	0.25 ± 0.01	0.24 ± 0.01	0.21 ± 0.04
Insulin sensitivity (M/I)	110–120 min	16 ± 3	18 ± 3	11 ± 2

Data are presented as mean ± SEM unless specified otherwise. GB, patients with prior history of gastric bypass; SG, sleeve gastrectomy patients; CN, non-surgical controls; GIR, glucose infusion rate; ISR, insulin secretion rate; ICR, insulin clearance; 1 h: 0–60 min after the meal; 3 h: 0–180 min after the meal; ***** *p* < 0.05 vs. surgical groups.

**Table 3 ijms-23-07667-t003:** Glucose and insulin profile during mixed-meal test.

		GB (n = 9)	SG (n = 7)
Glucose (mmol/L)	Fasting	5.4 ± 0.2	5.3 ± 0.2
Time to glucose peak (min)		20 ± 2	19 ± 6
AUC_Glucose_ (mmol·m^−2^)	0–60 min	251 ± 38	142 ± 23 *
	0–180 min	219 ± 57	193 ± 39
ISR (pmol·min^−1^·m^−2^)	Fasting	160 ± 27	101 ± 11
Time to ISR peak (min)		17 ± 3	17 ± 7
AUC_ISR_ (nmol·m^−2^)	0–60 min	2.5 ± 1.0	0.6 ± 0.2 *
	0–180 min	105 ± 39	36 ± 7 *
Insulin (pmol/L)	Fasting	74 ± 7	47 ± 8
Time to insulin peak (min)		31 ± 3	24 ± 7
AUC_Insulin_ (nmol·m^−2^·min^−1^)	0–60 min	68 ± 10	33 ± 9 *
	0–180 min	77 ± 10	48 ± 13
Hepatic insulin fractional extraction (%)		69 ± 3	70 ± 4
ICR (L·min^−1^·m^−2^)	Fasting	3.5 ± 0.5	3.4 ± 0.6
AUC_ICR_ (L·m^−2^)	0–60 min	−102 ± 22	−108 ± 28
	0–180 min	−317 ± 72	−282 ± 77

Data are presented as mean ± SEM unless specified otherwise. GB, patients with prior history of gastric bypass; SG, sleeve gastrectomy patients; ISR, insulin secretion rate; ICR, insulin clearance rate; 0–60 min after the meal; 0–180 min after the meal; ***** *p* < 0.05 vs. GB.

## Data Availability

All data generated or analyzed during this study are included in this published article.

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
