# Peer review of "Altered Insulin Clearance after Gastric Bypass and Sleeve Gastrectomy in the Fasting and Prandial Conditions"

_ijms, 2022, doi:10.3390/ijms23147667_

Round 1

Reviewer 1 Report

Thank you for the opportunity to review this interesting manuscript by Marzieh Salehi, Ralph DeFronzo and Amalia Gastaldelli. The study is of high international standard covering a very interesting topic and the manuscript is written by top expert in the field.

 Nevertheless, I have a couple of points to be better clarified in the manuscript.

1. Description of the models used to calculate insulin clearance/extraction rates is limited. Since it is the main finding of the study, the manuscript should provide a better description of the calculations of insulin clearance rates (including hepatic vs total)/ insulin extraction including assumptions and limitations. Moreover, it would be helpful for the reader to include definitions/descriptions of central terms. For instance it should be clearly stated that insulin extraction rates in this manuscript are the combined extraction rate of exogenous and endogenous insulin as opposed to only endogenous insulin.

2. Fractional insulin extraction is not estimated and only briefly discussed. Insulin extraction is calculated as the absolute rate of extraction and is not related to the rate of insulin secretion, which differs substantially between GB/SG/CON and is highly dependent on the settings of the meal test (30 fold lower during concomitant hypoglycemia/high concentrations of exogenous insulin). Hence, the importance of different insulin secretion rates for the findings of the study should be elaborated in the manuscript. 

3. Are there previous reports on the same cohort? If yes, this should be described in the manuscript.

4. There is no report on post-meal glucose concentrations during the hypoglycemic clamp, which should be shown. Is the higher insulin secretion in GB vs SG vs CON patients after meal ingestion during hypoglycemia caused by differences in glucose cocentrations in GB vs SG/CON? What about data on counterregulatory responses?

5. In figure 2, the insulin extraction rate is depicted as a function of systemic insulin concentrations. However, the insulin extraction rate and systemic insulin concentration are not independent variables as systemic insulin concentrations are very much dependent on the rate of insulin extraction. Please justify/comment.

6.  Peripheral glucose disposal rates (during hypoglycemia) are reported equal between groups (Table 2). The role of counterregulatory responses and inadequate power (7 vs 5) should be discussed. 

Minor comments:

Introduction l. 41-4.  Postprandial insulin clearance is reported be reduced in GB, which is not a general finding (see also reference 11).  Please correct.

Table 1: Add %excess BMI loss

Table 2 is devided between page 3 and 4, which makes it hard to read. 

Figure legend for figure 2 is placed both on page 4 and 5.

Line 258-260: Argumentation hard to follow. Should be elaborated.

Reviewer 2 Report

I have several comments to this article.

-Line 173, this study was not intervention study, thus, it is hard to say that “improved insulin clearance after GB and SG”.

-Again, this study is cross-sectional, and the number of CN subjects was very low (only 5). The CN subjects were relatively low lean mass and high fat mass compared with other groups. The group differences may be due to the difference of any kind of characteristics. The slope shown in Figure 2a was correlated to the characteristics shown in table 1 or other parameters? Anyway, authors need to carefully explain the results of the present study and revise through the manuscript. There is a large limitation.

-Insulin levels were once reduced after mixed meal during the clamp, and they were increased in 3 groups. Why this happened? How was the mixed meal provided during the clamp? Subjects consumed it by sitting position?

-Several data should be shown by figures. For example, GIR, glucose and C-peptide data for the clamp study should be shown by figures.

-Peripheral insulin sensitivity may not be calculated by reliable method. It was adjusted insulin concentration during steady state, but is there any rationale for this? The GIR could be increased in proportion to insulin concentration in such high insulin level? If insulin level is increased from 1400 to 2100, does the GIR increase by 1.5 times?

-The GIR should be adjusted by body weight, but fat free mass.

Round 2

Reviewer 2 Report

Well revised.